# Plasma Metabolites Associated with Brain MRI Measures of Neurodegeneration in Older Adults in the Atherosclerosis Risk in Communities–Neurocognitive Study (ARIC-NCS)

**DOI:** 10.3390/ijms20071744

**Published:** 2019-04-09

**Authors:** Danni Li, Jeffrey R. Misialek, Clifford R. Jack, Michelle M. Mielke, David Knopman, Rebecca Gottesman, Tom Mosley, Alvaro Alonso

**Affiliations:** 1Department of Lab Medicine and Pathology, University of Minnesota, 420 Delaware Street SE, MMC 609, Minneapolis, MN 55455, USA; 2Division of Epidemiology and Community Health, School of Public Health, University of Minnesota, Minneapolis, MN 55455, USA; misi0020@umn.edu; 3Department of Radiology, Mayo Clinic College of Medicine, Rochester, MN 55905, USA; jack.clifford@mayo.edu; 4Department of Health Sciences Research, Mayo Clinic College of Medicine, Rochester, MN 55906, USA; mielke.michelle@mayo.edu; 5Department of Neurology, Mayo Clinic College of Medicine, Rochester, MN 55905, USA; knopman@mayo.edu; 6Department of Neurology, Johns Hopkins Medical Institutions, Baltimore, MD 21287, USA; rgottesm@jhmi.edu; 7Department of Medicine, University of Mississippi Medical Center, Jackson, MS 39216, USA; tmosley@umc.edu; 8Department of Epidemiology, Rollins School of Public Health, Emory University, Atlanta, GA 30322, USA; alvaro.alonso@emory.edu

**Keywords:** metabolomics, sphingomyelins, plasma, cerebral small vessel disease, neurodegeneration, brain atrophy

## Abstract

Background: Plasma metabolites are associated with cognitive and physical function in the elderly. Because cerebral small vessel disease (SVD) and neurodegeneration are common causes of cognitive and physical function decline, the primary objective of this study was to investigate the associations of six plasma metabolites (two plasma phosphatidylcholines [PCs]: PC aa C36:5 and PC aa 36:6 and four sphingomyelins [SMs]: SM C26:0, SM [OH] C22:1, SM [OH] C22:2, SM [OH] C24:1) with magnetic resonance imaging (MRI) features of cerebral SVD and neurodegeneration in older adults. Methods: This study included 238 older adults in the Atherosclerosis Risk in Communities study at the fifth exam. Multiple linear regression was used to assess the association of each metabolite (log-transformed) in separate models with MRI measures except lacunar infarcts, for which binary logistic regression was used. Results: Higher concentrations of plasma PC aa C36:5 had adverse associations with MRI features of cerebral SVD (odds ratio of 1.69 [95% confidence interval: 1.01, 2.83] with lacunar infarct, and beta of 0.16 log [cm3] [0.02, 0.30] with log [White Matter Hyperintensities (WMH) volume]) while higher concentrations of 3 plasma SM (OH)s were associated with higher total brain volume (beta of 12.0 cm3 [5.5, 18.6], 11.8 cm3 [5.0, 18.6], and 7.3 cm3 [1.2, 13.5] for SM [OH] C22:1, SM [OH] C22:2, and SM [OH] C24:1, respectively). Conclusions: This study identified associations between certain plasma metabolites and brain MRI measures of SVD and neurodegeneration in older adults, particularly higher SM (OH) concentrations with higher total brain volume.

## 1. Introduction

Cerebral small vessel diseases (SVD) and neurodegeneration frequently coexist and are common causes of poor physical and cognitive function in the elderly [1]. Features of SVD and neurodegeneration seen on brain neuroimaging include lacunar infarcts (small recent subcortical infarcts), white matter hyperintensities (WMH), and brain atrophy [2]. Pathogeneses and neuropathological changes of these brain MRI features are not well-understood and could be multifactorial and heterogeneous, respectively [2,3]. Several lines of evidence suggest endothelial dysfunction, often manifested as inflammatory endothelial cell activation, damaged vascular permeability, or impaired cerebral blood flow and autoregulation, play a pivotal role in the pathogeneses of cerebral SVD and neurodegeneration [3]. Molecular components in plasma may modulate endothelial function and hence influence the pathogeneses of cerebral SVD and neurodegeneration [3,4]. Previous studies of plasma molecular components and neuroimaging features of SVD examined markers of inflammation (i.e., TGF-beta, C-reactive protein, interleukin-6, YKL-40) [1,5,6], endothelial dysfunction (i.e., intercellular adhesion molecule, thrombomodulin, tissue factor, and tissue factor pathway inhibitor, homocysteine) [3,4], and well-established lipid biomarkers of cardiovascular disease risk (triglycerides and cholesterol) [7]. However, these plasma molecular components fail to account for much of the risk of cerebral SVD or the neuroimaging features [3].

Plasma phospholipids and sphingolipids play important roles in vascular health and microvascular pathology such as endothelial dysfunction [8,9,10,11,12]. Because of the underlying mechanistic connections between plasma phospholipids and sphingolipids and endothelial dysfunction, the central hypothesis of this study is that plasma phospholipids and sphingolipids may be associated with neuroimaging features of cerebral SVD and neurodegeneration, the underlying mechanism of poor physical and cognitive function. Our previous studies of 188 plasma metabolites in the Atherosclerosis Risk in Communities (ARIC) study found that lower concentrations of two phosphatidylcholines (PCs: PC aa C36:5, PC aa C36:6), a subclass of phospholipids, and four sphingomyelins (SMs: SM C26:0, SM (OH) C22:1, SM (OH) C22:2, SM (OH) C24:1), a subclass of sphingolipids, were cross-sectionally associated with worse cognitive status and poor physical function in the elderly [13,14]. The objective of this study is to examine the plausible associations between six plasma phospholipids and sphingolipids (PC aa C36:5, PC aa C36:6, SM C26:0, SM [OH] C22:1, SM [OH] C22:2, and SM [OH] C24:1) and neuroimaging features of cerebral SVD (lacunar infarcts and WMH volume) and neurodegeneration (brain volume) in older adults. We hypothesize that lower concentrations of plasma PC aa C36:5, PC aa C36:6, SM C26:0, SM (OH) C22:1, SM (OH) C22:2, and SM (OH) C24:1 are associated with higher neuroimaging burden of cerebral SVD in older adults. Since cerebral SVD also frequently coexists with neurodegenerative disease such as Alzheimer’s disease (AD) [8], this study examines the associations of these six plasma metabolites with AD-associated neurodegeneration (total AD signature region cortical volume). In exploratory analyses, we examine 131 additional plasma metabolites with these SVD and neurodegeneration outcomes. 

## 2. Results

Table 1 lists the characteristics of the 238 ARIC visit 5 participants with both phospholipid measures and brain MRI measures who are included in this study, compared to the ARIC visit 5 participants who are excluded (203 participants with phospholipids measures but without brain MRI measures, and 1740 and 4560 ARIC participants without phospholipids measures either with and without brain MRI measures, respectively). The ARIC participants included in the present study had similar demographic characteristics than the excluded participants. Furthermore, the included ARIC participants had higher prevalence of lacunar infarcts and higher volume of WMH, compared to the excluded participants with brain MRI measures. However, there were no group differences in other brain MRI characteristics (total brain volume or total AD signature region volume). 

Among the six plasma metabolites examined in the primary hypothesis, PC aa C36:5 was the metabolite whose associations with both brain MRI measures of SVD mostly strengthened in Model 2 after adjustment of additional covariates, compared to Model 1 (age, race/center, sex, *APOE*, education level only): odds ratio with lacunar infarcts of 1.13 (95% confidence interval [CI]: 0.74, 1.72) in Model 1 versus 1.69 (1.01, 2.83) in Model 2 (Table 2); and β (95% CI) of log(WMH) volume 0.04 log (cm^3^) (-0.09, 0.18) in Model 1 versus 0.16 log (cm^3^) (0.02, 0.30) in Model 2 (Table 3). The covariates in Model 2 that were responsible for strengthening the relations were sport index, depression, and prevalence of stroke (data not shown). In these strengthened relations, higher PC aa C36:5 concentration was independently associated with adverse brain MRI measures of SVD (higher odds of lacunar infarct and higher WMH volume). 

Higher plasma SM (OH) concentrations were associated with favorable brain MRI measures of SVD (SM [OH] C24:1 and log(WMH) volume had β [95% CI] of −0.19 log (cm^3^) [−0.31, −0.06] and −0.21 log (cm^3^) [−0.33, −0.09] in Models 1 and 2, respectively, Table 3) and higher total brain volume (SM [OH] C22:1, SM [OH] C22:2, and SM [OH] C24:1 and total brain volume had β [95% CI] of 12.04 cm^3^ [5.53, 18.55], 11.79 cm^3^ [5.03, 18.56], and 7.33 cm^3^ [1.20, 13.46] in Model 2, respectively, Figure 1 and Table 4). We did not find any significant associations between any of these 6 plasma metabolites with total AD signature region cortical volume. Given that all three SM(OH)s demonstrated significant associations with total brain volume, we summed up their plasma levels (log transformed per 1-SD change) and analyzed as a single income. Results from this additional analysis demonstrated that Total SM(OH) was significantly associated with total brain volume (β [95% CI] of 6.95 cm^3^ [0.35, 13.54], 12.61 cm^3^ [6.36, 18.86] in Models 1 and 2, respectively, Figure 1 and Table 5). 

Our exploratory analyses examined cross-sectional relations of 131 additional plasma metabolites with brain MRI measures of SVD and AD (Appendix A). Higher concentrations of plasma PC aa C38:6 and PC aa C40:6 were adversely associated with lacunar infarct (odds ratios [95% CI] of 2.98 [1.68, 5.32] and 2.75 [1.65, 4.58] in Model 2, respectively), but not with log(WMH) volume. Higher concentrations of plasma arginine, glutamine, and PC ae C40:4, PC ae C42:4, and PC ae C42:5 were favorably associated with lower log(WMH) volume only (β [95% CI] of −0.20 log [cm^3^] [−0.30, −0.11], −0.22 log [cm^3^] [−0.32, −0.11], −0.25 log [cm^3^] [−0.38, −0.12], −0.29 cm^3^ [−0.42, −0.16], and −0.25 log [cm^3^] [−0.39, −0.12] in Model 2, respectively).

## 3. Discussion

This study identified independent favorable (3 SMs: SM [OH] C22:1, SM [OH] C22:2, SM [OH] C24:1; 3 PC ae: PC ae C40:4, PC ae C42:4, PC ae C42:5; arginine; glutamine) and adverse associations (3 PC aa: PC aa C36:5, PC aa C38:6, PC aa C40:6) between plasma metabolites and brain MRI measures of neurodegeneration in older adults. Similar to previous studies, many of these plasma metabolites correlated with the brain MRI measures of SVD and neurodegeneration modestly. However, the most meaningful findings were the striking magnitude of associations between higher SM (OH) concentrations with higher total brain volume (less brain atrophy). 

Sphingolipids are major lipids in the myelin sheath that are critical for the proper functioning of both the central and peripheral nervous system. SM (OH)s, hydroxyl derivatives of SMs, are particularly important for long-term stability of the myelin sheath, where their deficit leads to late onset demyelination and neurodegeneration [15,16]. This study built upon the previous established favorable associations between these three plasma SM (OH)s and cognitive [14] and physical function [13] by further linking them positively to brain morphology (total brain volume). The strong and positive correlations between higher plasma SM (OH)s and higher total brain volume (less brain atrophy) are consistent with the abundance of SM (OH)s detected in the brain [17]. 

Similarly, PC aa C36:5 had an adverse relation to brain MRI features of SVD, opposite to our hypothesis. However, the adverse relation was consistent with the exploratory findings, which showed higher concentrations of two other PCs, PC aa C38:6 and PC aa C40:6, also negatively associated with brain MRI features of SVD (odds of lacunar infarcts). One possible explanation of the adverse relation is that these PCs (PC aa C36:5, PC aa C38:6 and PC aa 40:6) contain polyunsaturated fatty acids that are prone to damage by reactive oxygen species including peroxidation, and is involved in many proinflammatory pathways [18] and chronic inflammation, a driving force in aging and disease [19]. The associations between PC aa C36:5 and brain MRI measures of SVD strengthened after adjustment for additional covariates in Model 2, in particular sport index, depression, and prevalence of stroke. Reasons for the stronger associations are unclear. In the regression output there was no evidence of overfitting and we consider this change may be due the complex relations between brain MRI measures of SVD and these three covariates.

The exploratory analysis showed higher levels of some PC ae species (PC ae C40:4, PC ae C42:4, and PC ae C42:5) favorably associated with lower WMH volume, consistent with their antioxidative properties [20] that are protective of endothelial dysfunction and hence brain MRI features of SVD. In addition, the study found favorable associations of higher concentrations of arginine and glutamine with lower WMH volume. Previous studies demonstrated that arginine and glutamine are the physiological substrates for the synthesis of nitric oxide [21], a vasodilator and mediator of cerebral blood flow and autoregulation; and oral arginine supplement improved endothelium-dependent dilation [22], an outcome that was inversely associated with WMH in older adults with cardiovascular disease [23]. Our finding builds upon this previous knowledge by establishing independent associations between lower WMH and concentrations of these two amino acids in plasma. 

Although lacunar infarcts, WMH, and brain atrophy are all considered as brain MRI measures of SVD, they reflect distinctive aspects of cerebral SVD and are not exchangeable [2]. Lacunar infarcts may be associated with occlusion of small arteries and may evolve into a lacunar cavity, or hyperintensities without apparent cavitation on T2-weighted sequence, or disappear; WMHs correspond to neuronal loss, demyelination, and gliosis; brain atrophy include neuronal loss, cortical thinning, subcortical vascular pathology with white matter rarefaction and shrinkage, arteriolosclerosis, and venous collagenosis, and secondary neurodegeneration. Because these brain MRI features of SVD represent diverse neuropathological changes, it is possible that the association between a plasma metabolite with one MRI feature (lacunar infarcts) may not be equivalent with a different MRI feature (WMH). 

The strength of this study was that it accounted for important confounders such as SVD risk factors (hypertension), medication uses, and other factors that can impact plasma levels of metabolites. For example, plasma SM concentrations are affected by age, sex, race, lifestyle factors, and diseases [24]. Despite the strength, this study has limitations. First, the cross-sectional study design cannot assess temporality of associations. Second, our study had approximately 4:4:3 ratio of participants with cognitive normal, MCI, and dementia, and did not represent the general aging population. For example, the expected ratio of older adults with cognitive normal, MCI and dementia in communities such as ARIC was approximately 14:4:1. Third, residual confounding could be present. Fourth, the study findings are preliminary and need to be further validated. Fifth, we acknowledge that lack of significant associations for PC aa C36:6 or other metabolites may be due to the lack of power. Sixth, we imputed missing metabolite levels with lowest nonzero measurement, which may introduce bias [25]. However, none of the six metabolites included in the primary analysis (PC aa C36:5, PC aa C36:6, SM C26:0, SM [OH] C22:1, SM [OH] C22:2, and SM [OH] C24:1) had any missing values. Lastly, adjustment for multiple comparisons was not considered for the six plasma metabolites with previously demonstrated relations with cognitive and physical function decline, which may be considered as a limitation. However, we do not believe multiple comparison tests were necessary for the six metabolites as the analysis was hypothesis-driven, not exploratory, and our primary focus was on estimation of associations, not statistical significance testing.

## 4. Materials and Methods

### 4.1. Population

The ARIC study is a prospective cohort investigating the etiology of atherosclerotic disease in a middle-aged, mostly biracial (black and white) population in four U.S. communities. The ARIC study and ARIC-Neurocognitive Study (ARIC-NCS) were approved by the Institutional Review Board of each participating center, and written informed consent was obtained from participants at each study visit. At the University of Minnesota, the ARIC-NCS study was approved with an IRB ID: 1005M81992 and last date of approval of 7/24/2019. A detailed study description of the ARIC study was published [26]. In 2011–2013, the ARIC study conducted the fifth examination (visit 5) in 6538 of 10,036 living participants (65% participation rate) in conjunction with the ARIC-NCS, which assessed cognitive function and adjudicated mild cognitive impairment (MCI) and dementia cases [27]. Of the 6538 visit 5 participants, 344 received a diagnosis of dementia, 1374 were diagnosed as MCI, and 4755 were considered cognitively normal (65 with unknown status or data missing). In addition, 1906 of the visit 5 participants underwent brain MRI scans. For the present analysis, we only included 238 of the 441 ARIC participants from our previous study that investigated the cross-sectional association of plasma metabolites with cognitive status [14]. We excluded 203 participants because these participants did not receive brain MRI scans due to a selection process occurred in the ARIC-NCS [28]. In the previous study, these 441 participants were sampled to have approximately equal numbers of dementia, MCI, and cognitively normal participants and equal numbers of whites and blacks [14]. Cognitively normal individuals were frequency matched with dementia/MCI participants by *APOE* genotype, age (above or below the median age in the dementia/MCI participants), sex, education, and study center. 

### 4.2. Brain MRI Measures

The ARIC-NCS performed brain MRI scans on 3 Tesla Siemens scanners for the dual purposes of obtaining quantitative imaging features for analysis and supplementing the clinical etiologic diagnoses by documenting cerebrovascular lesions. As previously described, lacunar infarcts (i.e., less than 20 mm) were identified, counted, and measured by a trained imaging technician and confirmed by radiologists [28]. MR images were viewed on video monitors using locally developed software. Each scan was rated by an experienced image analyst and confirmed by a neuroradiologist. We used accepted criteria (shape, location) to differentiate dilated peri vascular spaces vs infarcts.

WMH were defined per recent guideline [2]. Quantitative measures of WMH volume were derived from the axial FLAIR images. We used a semiautomated algorithm developed in-house [29]. T1-weighted images were segmented using SPM5 and aligned to FLAIR images, and corresponding brain masks were used to remove nonbrain tissue and voxels that had a high probability of being cortical gray matter. Candidate WMH voxels were clustered via connected-components. Clusters were removed if they were mostly located outside regions classified as white matter, consisted of a single isolated voxel, or contained no supra-threshold FLAIR voxels after blurring. Remaining clusters were then manually edited by trained image analysts to correct for WMH misclassifications. Hyperintense voxels associated with infarcts were manually marked and excluded from the total WMH volume measurement as they are attributed to distinct pathophysiologic processes. All analysis involving WMH included total intracranial volume as a covariate. 

Freesurfer (version 5.1, Martinos Center for Biomedical Imaging, Boston, MA, USA) [30] was used to calculate total brain volumes and regional cortical volumes, reported in cubic millimeters. For this analysis, we have converted the units to cubic centimeters for easier interpretation. Total brain volume adjusted for intracranial volume is a biomarker of neurodegeneration, although it is not specific for AD [31,32]. Total AD signature region cortical volume including hippocampus, para hippocampal gyrus, entorhinal cortex, inferior parietal lobule, precuneus, and cuneus is a biomarker of neurodegeneration that is specific for AD [33]. 

### 4.3. Measurement of Plasma Metabolites

Plasma metabolites were measured as part of a targeted metabolomics panel using the Biocrates Absolute-IDQ P180 kits (Biocrates, Life Science AG, Innsbruck, Austria), as described elsewhere [14]. Fasting, frozen never thawed plasma samples were processed per the manufacturer instructions and analyzed on a triple-quadrupole mass spectrometer (QTRAP 6500, AB Sciex, Framingham, MA, USA). Three levels of quality controls were included in each kit to monitor imprecisions (% coefficient of variation [CVs]). Means, standard deviations (SDs), and CVs in 7 kits (the number of kits needed for analysis of the 441 samples) were calculated, and CVs of majority of the metabolites (i.e., > 80%) were less than 20%. We excluded those metabolites that had very high CVs [e.g., > 30%], which left 131 plasma metabolites in the secondary analysis. Metabolite concentrations were quantified by the MetIQ software (Biocrates). 

### 4.4. Covariates

The covariates considered in this study included age, race/ethnicity (white or African American), sex, *APOE* genotype, educational level, estimated total intracranial volume, body mass index, diabetes mellitus, drinking status, smoking status, sports index, systolic blood pressure, use of antihypertensive medications (yes or no), use of statins (yes or no), depression, prevalent coronary heart disease, heart failure, stroke, total cholesterol, high density lipoprotein (HDL) cholesterol, and triglycerides. Education level, sex, and race were self-reported by the participants at the ARIC study baseline visit in 1987–1989. The genotyping of APOE polymorphisms was performed using the TaqMan assay (Applied Biosystems, Foster City, CA, USA) during the ARIC visit 3 in 1993–1995 [34]. Because the standard TaqMan assay can only detect up to two single-nucleotide polymorphisms per reaction, the *APOE* variants at codons 112 and 158 were detected separately. The data from these two codons were combined to generate the six *APOE* genotypes (e2/e2, e2/e3, e2/e4, e3/e3, e3/e4, e4/e4). The rest of covariate information below was obtained at the ARIC visit 5 in 2011-2013: smoking and drinking status was self-reported; participants were asked to bring their medications, which were then recorded; body mass index was defined as weight in kilograms divided by the square of height in meters measured with the participant wearing light clothing; a sports index for physical activity was calculated based on the number of times per month that participants engaged in vigorous, moderate, or light physical activity and scored 1–5 as previously described [35]; prevalent coronary heart disease, stroke and heart failure were defined according to published criteria [36,37]; prevalent diabetes was defined as a fasting blood glucose ≥ 126 mg/dL, nonfasting glucose ≥ 200 mg/dL, self-reported physician-diagnosis of diabetes or use of antidiabetic medication; blood levels of total cholesterol, HDL cholesterol, and triglycerides were measured using standard methods [38]. 

### 4.5. Statistical Analysis 

Metabolite concentrations were log transformed and modeled in SD units. Eleven metabolites had some 0 values, to which we imputed the lowest non-zero measurement for analysis. Overall, 28 metabolites had >80% missing or below limit of detection (LOD) or had a failure in model convergence and therefore were excluded (Appendix A). Ten metabolites with 50–80% missing or below LOD values were categorized into three groups: <LOD, ≥LOD to <median, and ≥median (Appendix A). WMH volume was log transformed as well. Multiple linear regression was used to assess the association of each metabolite in separate models with neuroimaging measures except lacunar infarcts; for lacunar infarcts, binary logistic regression was used. An initial model adjusted for age, race/center, sex, *APOE*, education level, and estimated total intracranial volume (in the log[WMH] analyses and brain volume MRI measures only). A subsequent model additionally adjusted for body mass index, diabetes mellitus, drinking status, smoking status, sports index, systolic blood pressure, use of antihypertensive medications, use of statins, depression, prevalent coronary heart disease, heart failure, or stroke at ARIC visit 5, total cholesterol, HDL cholesterol, and triglycerides. Missing values for the covariates were imputed carrying forward the last available value from a prior visit or creating a category for missing values (i.e., Last Observation Carried Forward). Overall, only 10.9% (*n* = 26) had values missing and imputed for a covariate from a previous visit. In a secondary analysis, we further investigated association of 131 plasma metabolites with each neuroimaging measure. With these131 metabolites, statistical significance was defined as *p*-value < 0.00038 using a Bonferroni correction. However, for the 6 metabolites [PC aa C36:5, PC aa C36:6, SM C26:0, SM (OH) C22:1, SM (OH) C22:2, and SM (OH) C24:1] included in the primary analysis, statistical significance was not adjusted for multiple comparison. In all models, we weighted the observations by the inverse of the probability of these 238 participants of being included in the study to balance participants according cognitive phenotypes (normal, MCI, dementia). We did not use weights derived for the original matched sample, since this was larger than the one included in the present analysis and, therefore, imbalances may occur after restricting the analysis to a smaller sample. These probabilities were calculated with a logistic regression model using inclusion in the study as the dependent variable and age, sex, race, education, study center, dementia/MCI diagnosis and *APOE* genotype as predictors. Robust variance estimators were calculated to consider the weighting. We used SAS v 9.3 (SAS Inc., Cary, NC, USA) for the statistical analysis. Upon request, the SAS code and ARIC data (i.e., through the ARIC Coordinating Center) can be made available. 

## 5. Conclusions

We identified several novel associations of plasma metabolites with lacunar infarcts, WMH, and brain atrophy, in particular plasma SM (OH)s in older adults. These findings warrant replications in other populations and biochemical methods to measure plasma metabolites to confirm the observed associations and to evaluate the roles that these plasma metabolites could play in the risk of cerebral SVD among the elderly. 

## Figures and Tables

**Figure 1 ijms-20-01744-f001:**
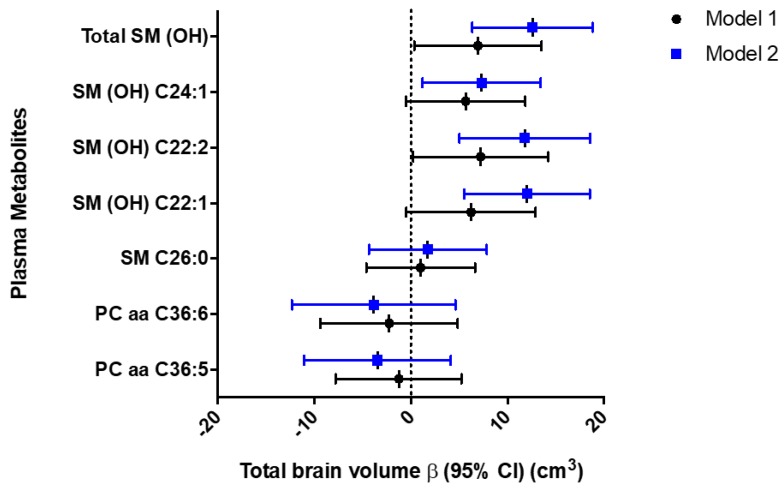
Forest plots of beta coefficients (95%CI) of multivariable linear regression analyses between 6 plasma metabolites (log-transformed per 1-SD change) and Total SM(OH) and total brain volume (cm^3^) in the ARIC study, 2011 to 2013. Model 1: Linear regression adjusted for age, race/center, sex, APOE, education level, and estimated total intracranial volume; Model 2: Model 1 with additional adjustment for body mass index, diabetes mellitus, drinking status, HDL cholesterol, smoking status, sports index, systolic blood pressure, total cholesterol, triglycerides, use of antihypertensive medications (yes or no), use of statins (yes or no), depression, and prevalence of coronary heart disease, heart failure, and stroke at the ARIC visit 5 baseline.

**Table 1 ijms-20-01744-t001:** Included and excluded Visit 5 characteristics of Atherosclerosis Risk in Communities (ARIC) Study participants (2011 to 2013).

		Participants Included	Participants Excluded		
			without Brain MRI but Metabolites	without Metabolites but Brain MRI	without Metabolites or Brain MRI
		(*n* = 238)	(*n* = 203)	(*n* = 1739)	(*n* = 4358)
Age, years		77.5 (5.6)	78.5 (5.5)	76.3 (5.3)	75.4 (5.2)
African American, %		21.9	36.0	29.4	20.8
Female, %		56.7	51.7	60.3	58.7
Body mass index, kg/m^2^		28.2 (5.4)	29.4 (6.1)	28.5 (5.7)	28.9 (5.9)
Current drinker, %		41.6	38.9	46.4	49.0
Current smoker, %		4.2	4.4	5.2	5.8
Diabetes, %		36.6	40.4	32.6	32.0
High school graduate or higher, %		83.2	74.9	85.8	85.0
HDL cholesterol, mg/dL		52.0 (14.8)	52.0 (14.0)	53.2 (14.3)	51.7 (13.9)
Hypertension medication, %		77.3	81.3	75.4	75.5
Statin medication, %		50.0	55.2	50.9	52.3
Prevalence of coronary heart disease, %		13.0	26.6	9.6	18.0
Prevalence of heart failure, %		4.6	13.8	5.0	10.0
Prevalence of stroke, %		3.8	8.9	3.4	4.3
Systolic blood pressure, mmHg		132.4 (19.3)	131.5 (19.2)	131.2 (18.4)	130.4 (18.7)
Sports index		2.6 (0.8)	2.5 (0.8)	2.6 (0.8)	2.6 (0.8)
Total cholesterol, mg/dL		180.8 (41.0)	181.4 (45.6)	183.9 (42.3)	181.1 (41.8)
Triglycerides, mg/dL		128.7 (75.8)	119.6 (54.8)	124.8 (62.1)	126.7 (64.9)
Depression score		3.1 (2.9)	3.8 (3.7)	3.2 (3.1)	3.1 (3.0)
APOE, %					
	e4/e4	3.4	4.9	2.4	1.9
	e2/e4 or e3/e4	31.9	33.0	24.8	25.5
	Other	62.6	57.6	69.0	68.2
	Missing	2.1	4.4	3.7	4.4
**Brain MRI characteristics**					
Lacunar infarcts, %		22.7	0	17.9	0
White matter hyperintensity volume, cm^3^		21.1 (20.2)	-	17.2 (16.7)	-
Total brain volume, cm^3^		1012.7 (108.2)	-	1011.5 (108.9)	-
Total AD signature region volume, cm^3^		58.6 (6.9)	-	58.9 (7.0)	-
Freesurfer deep grey cortical volume, cm^3^		42.6 (4.4)	-	42.6 (4.3)	-

Data are shown as percentage for categorical variables and as mean (standard deviation) for continuous variables. MCI = mild cognitive impairment; MRI = Magnetic resonance imaging; AD = Alzheimer’s disease. Variables below the dashed line are only present in the brain MRI subset study from visit 5.

**Table 2 ijms-20-01744-t002:** Weighted* cross-sectional associations of plasma metabolites with lacunar infarcts, Atherosclerosis Risk in Communities (ARIC) Study, 2011 to 2013. Odds ratios and 95% confidence intervals (CI) per 1-standard deviation difference in the log-transformed phospholipid.

Brain MRI	Metabolites	Model	n	Odds ratio (95% CI)	*p*-value
Lacunar infarct (*n* = 54)	PC aa C36:5	Model 1	238	1.13 (0.74, 1.72)	0.58
		Model 2	238	**1.69 (1.01, 2.83)**	**0.04**
	PC aa C36:6	Model 1	238	1.16 (0.76, 1.76)	0.49
		Model 2	238	1.48 (0.85, 2.58)	0.17
	SM C26:0	Model 1	238	1.03 (0.72, 1.46)	0.88
		Model 2	238	0.94 (0.52, 1.69)	0.84
	SM (OH) C22:1	Model 1	238	1.03 (0.67, 1.60)	0.88
		Model 2	238	0.85 (0.41, 1.74)	0.65
	SM (OH) C22:2	Model 1	238	0.98 (0.61, 1.57)	0.93
		Model 2	238	1.03 (0.52, 2.04)	0.94
	SM (OH) C24:1	Model 1	238	0.95 (0.65, 1.40)	0.80
		Model 2	238	0.74 (0.38, 1.46)	0.39
**Model 1**: Logistic regression adjusted for age, race/center, sex, APOE, and education level					
**Model 2**: Model 1 with additional adjustment for body mass index, diabetes mellitus, drinking status, HDL cholesterol, smoking status, sports index, systolic blood pressure, total cholesterol, triglycerides, use of antihypertensive medications (yes or no), use of statins (yes or no), depression, and prevalence of coronary heart disease, heart failure, and stroke at the ARIC visit 5.					

*****All analyses incorporated a weight accounting for selection into the metabolite study group from the ARIC visit 5 sample. Bold type indicates p value less than 0.05. MRI: Magnetic resonance imaging; WMH: White matter hyperintensity.

**Table 3 ijms-20-01744-t003:** Weighted* cross-sectional associations of plasma metabolites (log-transformed per 1-SD difference) with WMH (log-transformed), Atherosclerosis Risk in Communities (ARIC) Study, 2011 to 2013.

Brain MRI	Metabolites	Model	*n*	β (95% CI)^1^	*p*-value
**Log(WMH)**	PC aa C36:5	Model 1	237	0.04 (−0.09, 0.18)	0.54
		Model 2	237	**0.16 (0.02, 0.30)**	**0.02**
	PC aa C36:6	Model 1	237	0.04 (−0.12, 0.20)	0.62
		Model 2	237	0.14 (−0.02, 0.30)	0.08
	SM C26:0	Model 1	237	−0.06 (−0.20, 0.09)	0.46
		Model 2	237	−0.06 (−0.21, 0.09)	0.45
	SM (OH) C22:1	Model 1	237	−0.07 (−0.21, 0.07)	0.32
		Model 2	237	−0.14 (−0.30, 0.02)	0.09
	SM (OH) C22:2	Model 1	237	−0.01 (−0.17, 0.15)	0.94
		Model 2	237	−0.03 (−0.20, 0.15)	0.77
	SM (OH) C24:1	Model 1	237	**−0.19 (−0.31, −0.06)**	**0.003**
		Model 2	237	**−0.21 (−0.33, −0.09)**	**0.001**
**Model 1**: Linear regression adjusted for age, race/center, sex, APOE, education level, and estimated total intracranial volume					
**Model 2**: Model 1 with additional adjustment for body mass index, diabetes mellitus, drinking status, HDL cholesterol, smoking status, sports index, systolic blood pressure, total cholesterol, triglycerides, use of antihypertensive medications (yes or no), use of statins (yes or no), depression, and prevalence of coronary heart disease, heart failure, and stroke at the ARIC visit 5 baseline.					

*****All analyses incorporated a weight accounting for selection into the metabolite study group from the ARIC visit 5 sample.^1^ β (95% CI) corresponds to differences of log (WMH) in log (cm^3^) unit. Bold type indicates *p* value less than 0.05. MRI: Magnetic resonance imaging; WMH: White matter hyperintensity.

**Table 4 ijms-20-01744-t004:** Weighted* linear associations of plasma metabolites (log-transformed per 1-SD difference) with brain volume MRI measures, Atherosclerosis Risk in Communities (ARIC) Study, 2011 to 2013.

Brain MRI	Metabolites	Model	n	β (95% CI)^1^	*p*-value
Total brain volume	PC aa C36:5	Model 1	237	−1.26 (−7.78, 5.26)	0.70
		Model 2	237	−3.46 (−11.08, 4.15)	0.37
	PC aa C36:6	Model 1	237	−2.26 (−9.37, 4.86)	0.53
		Model 2	237	−3.86 (−12.33, 4.62)	0.37
	SM C26:0	Model 1	237	1.02 (−4.61, 6.65)	0.72
		Model 2	237	1.75 (−4.35, 7.85)	0.57
	SM (OH) C22:1	Model 1	237	6.23 (−0.46, 12.92)	0.07
		Model 2	237	**12.04 (5.53, 18.55)**	**0.0003**
	SM (OH) C22:2	Model 1	237	**7.23 (0.23, 14.24)**	**0.04**
		Model 2	237	**11.79 (5.03, 18.56)**	**0.001**
	SM (OH) C24:1	Model 1	237	5.67 (−0.51, 11.85)	0.07
		Model 2	237	**7.33 (1.20, 13.46)**	**0.02**
Total AD signature region cortical volume	**Phospholipid**	**Model**	**n**	**β (95% CI)**	***p*-value**
	PC aa C36:5	Model 1	236	0.07 (−0.50, 0.64)	0.82
		Model 2	236	−0.07 (−0.75, 0.61)	0.84
	PC aa C36:6	Model 1	236	0.41 (−0.17, 0.99)	0.16
		Model 2	236	0.32 (−0.41, 1.04)	0.39
	SM C26:0	Model 1	236	−0.36 (−0.93, 0.21)	0.22
		Model 2	236	−0.47 (−0.98, 0.04)	0.07
	SM (OH) C22:1	Model 1	236	0.19 (−0.46, 0.83)	0.57
		Model 2	236	−0.18 (−0.91, 0.56)	0.64
	SM (OH) C22:2	Model 1	236	0.28 (−0.41, 0.96)	0.43
		Model 2	236	0.01 (−0.72, 0.73)	0.99
	SM (OH) C24:1	Model 1	236	0.02 (−0.63, 0.66)	0.96
		Model 2	236	−0.30 (−0.98, 0.38)	0.39
**Model 1**: Linear regression adjusted for age, race/center, sex, APOE, education level, and estimated total intracranial volume.					
**Model 2**: Model 1 with additional adjustment for body mass index, diabetes mellitus, drinking status, HDL cholesterol, smoking status, sports index, systolic blood pressure, total cholesterol, triglycerides, use of antihypertensive medications (yes or no), use of statins (yes or no), depression, prevalence of coronary heart disease, heart failure, and stroke at the ARIC visit 5 baseline.					

*****All analyses incorporated a weight accounting for selection into the metabolite study group from the ARIC visit 5 sample.^1^ β (95% CI) corresponds to differences of brain volume MRI measures in cm^3^ unit. Bold type indicates p value less than 0.05. MRI: Magnetic resonance imaging; AD: Alzheimer’s disease.

**Table 5 ijms-20-01744-t005:** Weighted* linear associations of Total SM(OH) (log-transformed per 1-SD difference) with total brain volume, Atherosclerosis Risk in Communities (ARIC) Study, 2011 to 2013.

Brain MRI	Phospholipid	Model	*n*	β (95% CI)^1^	*p*-value
Total brain volume					
	Total SM(OH)	Model 1	237	**6.95 (0.35, 13.54)**	**0.04**
		Model 2	237	**12.61 (6.36, 18.86)**	**<0.0001**
**Model 1**: Linear regression adjusted for age, race/center, sex, APOE, education level, and estimated total intracranial volume.					
**Model 2**: Model 1 with additional adjustment for body mass index, diabetes mellitus, drinking status, HDL cholesterol, smoking status, sports index, systolic blood pressure, total cholesterol, triglycerides, use of antihypertensive medications (yes or no), use of statins (yes or no), depression, and prevalence coronary heart disease, heart failure, and stroke at the ARIC visit 5 baseline.					
All analyses incorporated a weight accounting for selection into the metabolite study group from the ARIC visit 5 sample.					
**Boldness indicates *p*-value less than 0.05.**					
MRI: Magnetic resonance imaging					

^1^ β (95% CI) corresponds to differences of brain volume MRI measures in cm^3^ unit.

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
