# Peer review of "Plasma Metabolites Associated with Brain MRI Measures of Neurodegeneration in Older Adults in the Atherosclerosis Risk in Communities–Neurocognitive Study (ARIC-NCS)"

_ijms, 2019, doi:10.3390/ijms20071744_

Reviewer 1 Report

Review for Danni Li et al. “ Plasma Metabolites Associated with Brain MRI 2 Measures in Older Adults in the Atherosclerosis Risk 3 in Communities – Neurocognitive Study (ARIC-NCS)”

This manuscript describes results showing some preliminary associations of plasma metabolites with MRI features of SVD and total brain volume. The authors investigate in particular phospholipids and sphingolipids in a hypothesis-driven approach but include also additional metabolites analysed in a hypothesis-free approach. The study gives the impression of being adapted already to previous reviewer comments.

My only major concern is the description of the MRI phenotypes measured, which I found very confusing:

-          There is no references supporting that brain volume measured by MRI is a surrogate marker for neurodegeneration.

-          The same is true for “AD-associated neurodegeneration” described as “total AD signature region cortical volume”. On which evidence is this phenotype description/assessment based?

-          Total AD signature region volume in Table 4 is not described in the results section. Since no significant findings could be achieved, this data can be eliminated from Table 4.

-          In the discussion section authors are overall stating to have used “Brain MRI measures of SVD and AD”: what exactly are they referring to for AD marker? In the introduction authors talk more generally about markers for neurodegeneration.

I understand that the selection of the two phosphatidylcholines and four sphingomyelins is based on previous studies but for me it would make more sense to assess all available phospholipids and sphingolipids in the hypothesis-driven approach, while the rest of the metabolites can be assessed in the hypothesis-free approach.

Furthermore:

This sentence in the introduction needs to be rephrased as it is not clear what the authors want to say with “…because of the underlying mechanisms…”.

“The central hypothesis of this study is that plasma phospholipids and sphingolipids that are associated with poor physical and cognitive function may also be associated with neuroimaging features of cerebral SVD and neurodegeneration because of the underlying mechanisms between plasma phospholipids and sphingolipids, endothelial dysfunction, cerebral SVD and neurodegeneration, and poor physical and cognitive function in older adults.”

The APOE genotype assessment and the explanation of the different genotypes should be given in the methods section.

These abbreviations should be explained: straight AP, LR and SI axes as well as central GM

Column description in Table 1 should be modified /shortened. In the description of Table 1 a “dashed line” is described, which is not present in this table.

The term “plasmalogen” needs to be explained or substituted by phospholipids.

On line 287 authors say: …and is  involved in the production of many pro-inflammatory eicosanoids…. What is meant here?

Author contributions are missing.

Author Response

This manuscript describes results showing some preliminary associations of plasma metabolites with MRI features of SVD and total brain volume. The authors investigate in particular phospholipids and sphingolipids in a hypothesis-driven approach but include also additional metabolites analysed in a hypothesis-free approach. The study gives the impression of being adapted already to previous reviewer comments.

My only major concern is the description of the MRI phenotypes measured, which I found very confusing:

-          There is no references supporting that brain volume measured by MRI is a surrogate marker for neurodegeneration.

Response:  We included references 31 and 32 to support total brain volume measured by MRI is a non-specific marker for neurodegeneration. This is because brain atrophy plagues more than just people with neurodegenerative disorders. Even seemingly healthy individuals lose brain matter over time.

31.       Erten-Lyons D, Dodge HH, Woltjer R, et al. Neuropathologic basis of age-associated brain atrophy. JAMA Neurology. 2013;70(5):616-622.

32.     Fjell AM, McEvoy L, Holland D, Dale AM, Walhovd KB, Initiative AsDN. Brain changes in older adults at very low risk for Alzheimer's disease. Journal of Neuroscience. 2013;33(19):8237-8242.

-          The same is true for “AD-associated neurodegeneration” described as “total AD signature region cortical volume”. On which evidence is this phenotype description/assessment based?

Response: We included reference 33 to support total AD signature region cortical volume including hippocampus, para hippocampal gyrus, entorhinal cortex, inferior parietal lobule, precuneus, and cuneus is a biomarker of neurodegeneration that is specific for AD.

33: Schwarz CG, Gunter JL, Wiste HJ, et al. A large-scale comparison of cortical thickness and volume methods for measuring Alzheimer's disease severity. NeuroImage: Clinical. 2016;11:802-812.

-          Total AD signature region volume in Table 4 is not described in the results section. Since no significant findings could be achieved, this data can be eliminated from Table 4.

Response: Even the data was not significant, We decided to leave these results in the table, for completeness purposes. We did comment in the results section that we did not find any significant associations.  Total AD signature region volume in Table 4.

-          In the discussion section authors are overall stating to have used “Brain MRI measures of SVD and AD”: what exactly are they referring to for AD marker? In the introduction authors talk more generally about markers for neurodegeneration.

Response: We agreed with the reviewer with the overstatement. We did not find significant associations between any metabolite and total AD signature region cortical volume, which we referred to as an AD neurodegeneration marker. Therefore, we revised this sentence in Discussion (lines 265-266) from "brain MRI measures of SVD and AD" to "brain MRI measures of neurodegeneration".

I understand that the selection of the two phosphatidylcholines and four sphingomyelins is based on previous studies but for me it would make more sense to assess all available phospholipids and sphingolipids in the hypothesis-driven approach, while the rest of the metabolites can be assessed in the hypothesis-free approach.

Response: We decided to assess the two phosphatidylcholines and four sphingomyelins in the hypothesis-driven approach, because they were based on previous studies. We discussed all other metabolites including other phospholipids and sphingolipids in the in the hypothesis-free approach. We believe this approach keeps our analysis more focused and avoids the risk of false positives.

Furthermore:

This sentence in the introduction needs to be rephrased as it is not clear what the authors want to say with “…because of the underlying mechanisms…”.

“The central hypothesis of this study is that plasma phospholipids and sphingolipids that are associated with poor physical and cognitive function may also be associated with neuroimaging features of cerebral SVD and neurodegeneration because of the underlying mechanisms between plasma phospholipids and sphingolipids, endothelial dysfunction, cerebral SVD and neurodegeneration, and poor physical and cognitive function in older adults.”

 Response: We have revised the sentence as follow in lines 65-69:

"Because of the underlying mechanistic connections between plasma phospholipids and sphingolipids and endothelial dysfunction, the central hypothesis of this study is that plasma phospholipids and sphingolipids may be associated with neuroimaging features of cerebral SVD and neurodegeneration, the underlying mechanism of poor physical and cognitive function."

The APOE genotype assessment and the explanation of the different genotypes should be given in the methods section.

Response: We provided the information on the APOE genotype assessment and the explanation of the different genotypes in the methods section in lines 409-412.

These abbreviations should be explained: straight AP, LR and SI axes as well as central GM

Response: We removed this sentence that contained these abbreviations as they are not relevant to the study.

Column description in Table 1 should be modified /shortened. In the description of Table 1 a “dashed line” is described, which is not present in this table.

Response: We shortened the column descriptions in Table 1. The dashed line is present in Table 1.

The term “plasmalogen” needs to be explained or substituted by phospholipids.

Response: We substituted plasmalogens with PCs.

On line 287 authors say: …and is involved in the production of many pro-inflammatory eicosanoids…. What is meant here?

Reponses: We clarified the sentence as follows:

"The exploratory analysis showed higher levels of some PC ae species (PC ae C40:4, PC ae C42:4, and PC ae C42:5) favorably associated with lower WMH volume, consistent with their anti-oxidative properties 20 that are protective of endothelial dysfunction and hence brain MRI features of SVD."

Author contributions are missing.

Response: we have added author contributions.

Reviewer 2 Report

The main aim of this study was to investigate in 238 older adults, included in the Atherosclerosis Risk in Communities study, the associations of six plasma metabolites  with magnetic resonance imaging features of cerebral small vessel disease and neurodegeneration. Normally multiple linear regression was used to assess the association of each metabolite  in separate models with magnetic resonance imaging features. Only for lacunar infarcts the binary logistic regression was used. The authors demonstrated the association between some metabolites and the measures of magnetic resonance imaging features. 

The study is quite interesting and is clear. However they need to better clarify  they found only associations, but they are not able to give a contribution about the possible pathogenetic role of studied metabollities.

Author Response

The main aim of this study was to investigate in 238 older adults, included in the Atherosclerosis Risk in Communities study, the associations of six plasma metabolites with magnetic resonance imaging features of cerebral small vessel disease and neurodegeneration. Normally multiple linear regression was used to assess the association of each metabolite in separate models with magnetic resonance imaging features. Only for lacunar infarcts the binary logistic regression was used. The authors demonstrated the association between some metabolites and the measures of magnetic resonance imaging features. 

The study is quite interesting and is clear. However, they need to better clarify they found only associations, but they are not able to give a contribution about the possible pathogenetic role of studied metabolites.

Response: We deleted a sentence that gave a contribution about the possible pathogenetic role of studied metabolites in Discussion.

"Although further studies remain to examine the causal relationship between plasma SM(OH)s and brain atrophy, a direct consequence of this present study findings is to support future examination of plasma SM (OH)s as novel plasma biomarkers of demyelination and neurodegeneration for predicting brain atrophy in older adults."

Round  2

Reviewer 1 Report

The manuscript has improved in many aspects. I’m still not fully convinced about the inclusion of only 2 phospholipids in the hypothesis-driven approach (based on their previous description), but it does not change/affect the overall results and therefore it is ok.